# Robust Classification Under Sample Selection Bias

**Anqi Liu**
Department of Computer Science
University of Illinois at Chicago
Chicago, IL 60607
`aliu33@uic.edu`

**Brian D. Ziebart**
Department of Computer Science
University of Illinois at Chicago
Chicago, IL 60607
`bziebart@uic.edu`

## Abstract

In many important machine learning applications, the source distribution used to estimate a probabilistic classifier differs from the target distribution on which the classifier will be used to make predictions. Due to its asymptotic properties, sample reweighted empirical loss minimization is a commonly employed technique to deal with this difference. However, given finite amounts of labeled source data, this technique suffers from significant estimation errors in settings with large sample selection bias. We develop a framework for learning a robust bias-aware (RBA) probabilistic classifier that adapts to different sample selection biases using a minimax estimation formulation. Our approach requires only accurate estimates of statistics under the source distribution and is otherwise as robust as possible to unknown properties of the conditional label distribution, except when explicit generalization assumptions are incorporated. We demonstrate the behavior and effectiveness of our approach on binary classification tasks.

## 1 Introduction

The goal of supervised machine learning is to use available source data to make predictions with the smallest possible error (loss) on unlabeled target data. The vast majority of supervised learning techniques assume that source (training) data and target (testing) data are drawn from the same distribution over pairs of example inputs and labels, $P(x, y)$, from which the conditional label distribution, $P(y|x)$, is estimated as $\hat{P}(y|x)$. In other words, data is assumed to be *independent and identically distributed* (IID). For many machine learning applications, this assumption is not valid; e.g., survey response rates may vary by individuals' characteristics, medical results may only be available from a non-representative demographic sample, or dataset labels may have been solicited using active learning. These examples correspond to the covariate shift [1] or missing at random [2] setting where the source dataset distribution for training a classifier and the target dataset distribution on which the classifier is to be evaluated depend on the example input values, $x$, but not the labels, $y$ [1]. Despite the source data distribution, $P(y|x)P_{\text{src}}(x)$, and the target data distribution, $P(y|x)P_{\text{trg}}(x)$, sharing a common conditional label probability distribution, $P(y|x)$, all (probabilistic) classifiers, $\hat{P}(y|x)$, are vulnerable to sample selection bias when the target data and the inductive bias of the classifier trained from source data samples, $\tilde{P}_{\text{src}}(x)\tilde{P}(y|x)$, do not match [3].

We propose a novel approach to classification that embraces the uncertainty resulting from sample selection bias by producing predictions that are explicitly robust to it. Our approach, based on minimax robust estimation [4, 5], departs from the traditional statistics perspective by prescribing (rather than assuming) a parametric distribution that, apart from matching known distribution statistics, is the worst-case distribution possible for a given loss function. We use this approach to derive the *robust bias-aware (RBA) probabilistic classifier*. It robustly minimizes the logarithmic loss (logloss) of the target prediction task subject to known properties of data from the source distribution. The parameters of the classifier are optimized via convex optimization to match statistical properties

measured from the source distribution. These statistics can be measured without the inaccuracies introduced from estimating their relevance to the target distribution [1]. Our formulation requires any assumptions of statistical properties generalizing beyond the source distribution to be explicitly incorporated into the classifier's construction. We show that the prevalent importance weighting approach to covariate shift [1], which minimizes a sample reweighted logloss, is a special case of our approach for a particularly strong assumption: that source statistics fully generalize to the target distribution. We apply our robust classification approach on synthetic and UCI binary classification datasets [6] to compare its performance against sample reweighted approaches for learning under sample selection bias.

## 2 Background and Related Work

Under the classical statistics perspective, a parametric model for the conditional label distribution, denoted $\hat{P}_\theta(y|x)$, is first chosen (e.g., the logistic regression model), and then model parameters are estimated to minimize prediction loss on target data. When source and target data are drawn from the same distribution, minimizing loss on samples of source data, $\tilde{P}_{\text{src}}(x)\tilde{P}(y|x)$,

$$\underset{\theta}{\operatorname{argmin}}\, \mathbb{E}_{\tilde{P}_{\text{src}}(x)\tilde{P}(y|x)}[\text{loss}(\hat{P}_\theta(Y|X), Y)], \tag{1}$$

efficiently converges to the target distribution ($P_{\text{trg}}(x)P(y|x)$) loss minimizer. Unfortunately, minimizing the sample loss (1) when source and target distributions differ does not converge to the target loss minimizer. A preferred approach for dealing with this discrepancy is to use importance weighting to estimate the prediction loss under the target distribution by reweighting the source samples according to the target-source density ratio, $P_{\text{trg}}(x)/P_{\text{src}}(x)$ [1, 7]. We call this approach sample reweighted loss minimization, or the sample reweighted approach for short in our discussion in this paper. Machine learning research has primarily investigated sample selection bias from this perspective, with various techniques for estimating the density ratio including kernel density estimation [1], discriminative estimation [8], Kullback-Leibler importance estimation [9], kernel mean matching [10, 11], maximum entropy methods [12], and minimax optimization [13]. Despite asymptotic guarantees of minimizing target distribution loss [1] (assuming $P_{\text{trg}}(x) > 0 \implies P_{\text{src}}(x) > 0$),

$$\mathbb{E}_{P_{\text{trg}}(x)P(y|x)}[\text{loss}(\hat{P}_\theta(Y|X), Y)] = \lim_{n\to\infty} \underbrace{\mathbb{E}_{\tilde{P}_{\text{src}}^{(n)}(x)\tilde{P}(y|x)}\left[\frac{P_{\text{trg}}(X)}{P_{\text{src}}(X)}\text{loss}(\hat{P}_\theta(Y|X), Y)\right]}_{\text{Sample reweighted objective function}}, \tag{2}$$

sample reweighting is often extremely inaccurate for finite sample datasets, $\tilde{P}_{\text{src}}(x)$, when sample selection bias is large [14]. The reweighted loss (2) will often be dominated by a small number of datapoints with large importance weights (Figure 1). Minimizing loss primarily on these datapoints often leads to target predictions with overly optimistic confidence. Additionally, the specific datapoints with large importance weights vary greatly between random source samples, often leading to high variance model estimates. Formal theoretical limitations match these described shortcomings; generalization bounds on learning under sample selection bias using importance weighting have only been established when the first moment of sampled importance weights is bounded, $\mathbb{E}_{P_{\text{trg}}(x)}[P_{\text{trg}}(X)/P_{\text{src}}(X)] < \infty$ [14],

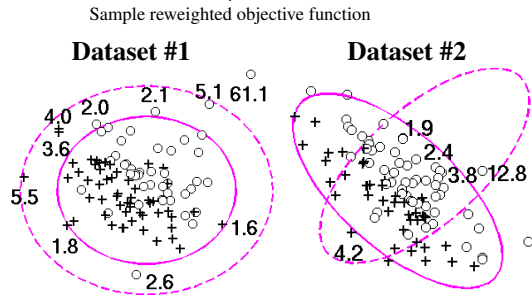

Figure 1: Datapoints (with '+' and 'o' labels) from two source distributions (Gaussians with solid 95% confidence ovals) and the largest data point importance weights, $P_{\text{trg}}(x)/P_{\text{src}}(x)$, under the target distributions (Gaussian with dashed 95% confidence ovals).

which imposes strong restrictions on the source and target distributions. For example, neither pair of distributions in Figure 1 satisfies this bound because the target distribution has "fatter tails" than the source distribution in some or all directions.

Though developed using similar tools, previous minimax formulations of learning under sample selection bias [15, 13] differ substantially from our approach. They consider the target distribution as being unknown and provide robustness to its worst-case assignment. The class of target distributions considered are those obtained by deleting a subset of measured statistics [15] or all possible

reweightings of the sample source data [13]. Our approach, in contrast, obtains an estimate for each given target distribution that is robust to all the conditional label distributions matching source statistics. While having an exact or well-estimated target distribution a priori may not be possible for some applications, large amounts of unlabeled data enable this in many batch learning settings.

A wide range of approaches for learning under sample selection bias and transfer learning leverage additional assumptions or knowledge to improve predictions [16]. For example, a simple, but effective approach to domain adaptation [17] leverages some labeled target data to learn some relationships that generalize across source and target datasets. Another recent method assumes that source and target data are generated from mixtures of "domains" and uses a learned mixture model to make predictions of target data based on more similar source data [18].

## 3  Robust Bias-Aware Approach

We propose a novel approach for learning under sample selection bias that embraces the uncertainty inherent from shifted data by making predictions that are explicitly robust to it. This section mathematically formulates this motivating idea.

### 3.1  Minimax robust estimation formulation

Minimax robust estimation [4, 5] advocates for the worst case to be assumed about any unknown characteristics of a probability distribution. This provides a strong rationale for maximum entropy estimation methods [19] from which many familiar exponential family distributions (e.g., Gaussian, exponential, Laplacian, logistic regression, conditional random fields [20]) result by robustly minimizing logloss subject to constraints incorporating various known statistics [21].

Probabilistic classification performance is measured by the conditional logloss (the negative conditional likelihood), $logloss_{P_{trg}(X)}(P(Y|X), \hat{P}(Y|X)) \triangleq \mathbb{E}_{P_{trg}(x)P(y|x)}[-\log P(Y|X)]$, of the estimator, $\hat{P}(Y|X)$, under an evaluation distribution (i.e., the target distribution, $P_{trg}(X)P(Y|X)$, for the sample selection bias setting). We assume that a set of statistics, denoted as convex set $\Xi$, characterize the source distribution, $P_{src}(x,y)$. Using this loss function, Definition 1 forms a robust minimax estimate [4, 5] of the conditional label distribution, $\hat{P}(Y|X)$, using a worst-case conditional label distribution, $\check{P}(Y|X)$.

**Definition 1.** *The **robust bias-aware (RBA) probabilistic classifier** is the saddle point solution of:*

$$\min_{\hat{P}(Y|X)\in\Delta} \max_{\check{P}(Y|X)\in\Delta \,\cap\, \Xi} logloss_{P_{trg}(X)}\left(\check{P}(Y|X), \hat{P}(Y|X)\right),\tag{3}$$

*where $\Delta$ is the conditional probability simplex: $\forall x \in \mathcal{X}, y \in \mathcal{Y} : P(y|x) \geq 0; \sum_{y'\in\mathcal{Y}} P(y'|x) = 1$.*

This formulation can be interpreted as a two-player game [5] in which the *estimator player* first chooses $\hat{P}(Y|X)$ to minimize the conditional logloss and then the *evaluation player* chooses distribution $\check{P}(Y|X)$ from the set of statistic-matching conditional label distributions to maximize conditional logloss. This minimax game reduces to a maximum conditional entropy [19] problem:

**Theorem 1** ([5]). *Assuming $\Xi$ is a set of moment-matching constraints, $\mathbb{E}_{P_{src}(x)\hat{P}(y|x)}[\mathbf{f}(X,Y)] = \mathbf{c} \triangleq \mathbb{E}_{P_{src}(x)P(y|x)}[\mathbf{f}(X,Y)]$, the solution of the minimax logloss game (3) maximizes the target distribution conditional entropy subject to matching statistics on the source distribution:*

$$\max_{\hat{P}(Y|X)\in\Delta} H_{P_{trg}(x),\hat{P}(y|x)}(Y|X) \text{ such that: } \mathbb{E}_{P_{src}(x)\hat{P}(y|x)}[\mathbf{f}(X,Y)] = \mathbf{c}.\tag{4}$$

Conceptually, the solution to this optimization (4) has low certainty where the target density is high by matching the source distribution statistics primarily where the target density is low.

### 3.2  Parametric form of the RBA classifier

Using tools from convex optimization [22], the solution to the dual of our constrained optimization problem (4) has a parametric form (Theorem 2) with Lagrange multiplier parameters, $\theta$, weighing

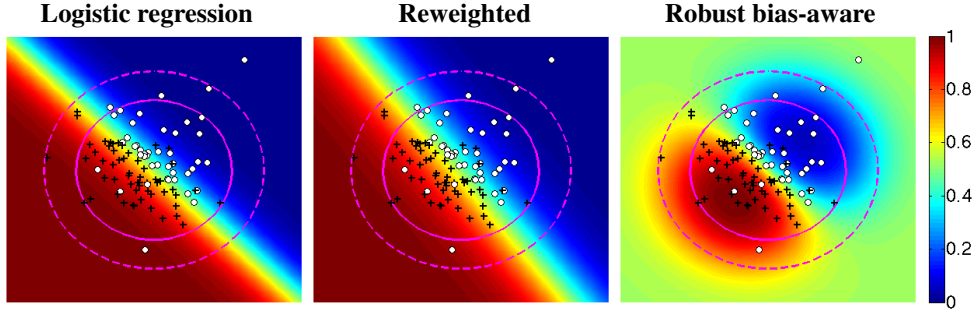

Figure 2: Probabilistic predictions from logistic regression, sample reweighted logloss minimization, and robust bias-aware models (§4.1) given labeled data ('+' and 'o' classes) sampled from the source distribution (solid oval indicating Gaussian covariance) and a target distribution (dashed oval Gaussian covariance) for first-order moment statistics (i.e., $\mathbf{f}(x,y) = [y\ yx_1\ yx_2]^T$).

the feature functions, $\mathbf{f}(x,y)$, that constrain the conditional label distribution estimate (4) (derivation in Appendix A). The density ratio, $P_{\mathrm{src}}(x)/P_{\mathrm{trg}}(x)$, scales the distribution's prediction certainty to increase when the ratio is large and decrease when it is small.

**Theorem 2.** *The robust bias-aware (RBA) classifier for target distribution $P_{src}(x)$ estimated from statistics of source distribution $P_{src}(x)$ has a form:*

$$\hat{P}_\theta(y|x) = \frac{e^{\frac{P_{src}(x)}{P_{trg}(x)}\theta\cdot\mathbf{f}(x,y)}}{\sum_{y'\in\mathcal{Y}} e^{\frac{P_{src}(x)}{P_{trg}(x)}\theta\cdot\mathbf{f}(x,y')}}, \tag{5}$$

*which is parameterized by Lagrange multipliers $\theta$. The Lagrangian dual optimization problem selects these parameters to maximize the target distribution log likelihood:* $\max_\theta \mathbb{E}_{P_{trg}(x)P(y|x)}[\log \hat{P}_\theta(Y|X)]$.

Unlike the sample reweighting approach, our approach does not require that target distribution support implies source distribution support (i.e., $P_{\mathrm{trg}}(x) > 0 \implies P_{\mathrm{src}}(x) > 0$ is not required). Where target support vanishes (i.e., $P_{\mathrm{trg}}(x) \to 0$), the classifier's prediction is extremely certain, and where source support vanishes (i.e., $P_{\mathrm{src}}(x) = 0$), the classifier's prediction is a uniform distribution. The critical difference in addressing sample selection bias is illustrated in Figure 2. Logistic regression and sample reweighted loss minimization (2) *extrapolate in the face of uncertainty* to make strong predictions without sufficient supporting evidence, while the RBA approach is *robust to uncertainty* that is inherent when learning from finite shifted data samples. In this example, prediction uncertainty is large at all tail fringes of the source distribution for the robust approach. In contrast, there is a high degree of certainty for both the logistic regression and sample reweighted approaches in portions of those regions (e.g., the bottom left and top right). This is due to the strong inductive biases of those approaches being applied to portions of the input space where there is sparse evidence to support them. The conceptual argument against this strong inductive generalization is that the labels of datapoints in these tail fringe regions could take either value and negligibly affect the source distribution statistics. Given this ambiguity, the robust approach suggests much more agnostic predictions.

The choice of statistics, $\mathbf{f}(x,y)$ (also known as features), employed in the model plays a much different role in the RBA approach than in traditional IID learning methods. Rather than determining the manner in which the model generalizes, as in logistic regression, features should be chosen that prevent the robust model from "pushing" all of its certainty away from the target distribution. This is illustrated in Figure 3. With only first moment constraints, the predictions in the denser portions of the target distribution have fairly high uncertainty under the RBA method. The larger number of constraints enforced by the second-order mixed moment statistics preserve more of the original distribution using the RBA predictions, leading to higher certainty in those target regions.

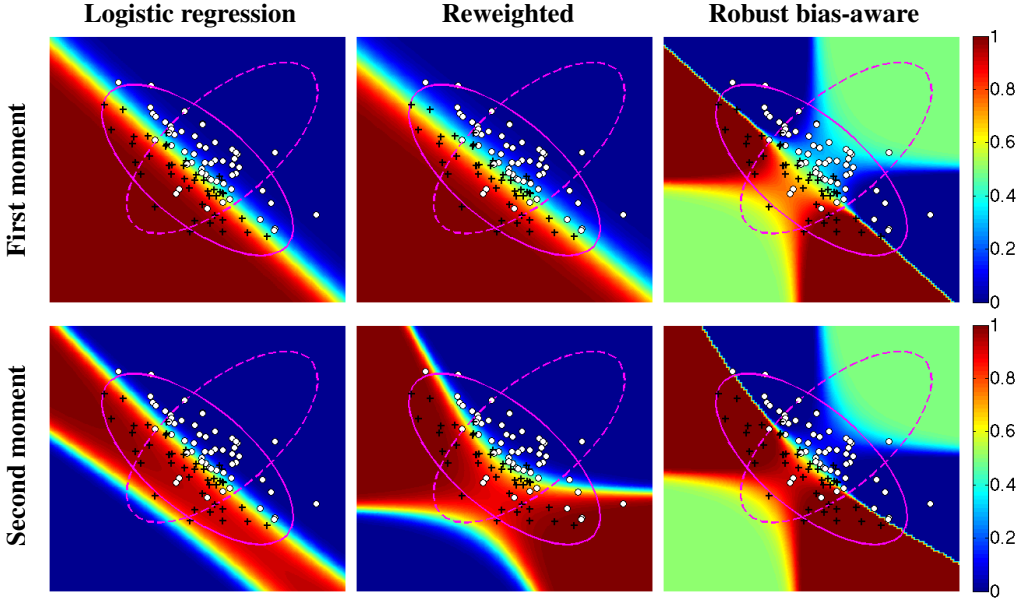

Figure 3: The prediction setting of Figure 2 with partially overlapping source and target densities for first-order (top) and second-order (bottom) mixed-moments statistics (i.e., $\mathbf{f}(x,y) = [y \; yx_1 \; yx_2 \; yx_1^2 \; yx_1x_2 \; yx_2^2]^T$). Logistic regression and the sample reweighted approach make high-certainty predictions in portions of the input space that have high target density. These predictions are made despite the sparseness of sampled source data in those regions (e.g., the upper-right portion of the target distribution). In contrast, the robust approach "pushes" its more certain predictions to areas where the target density is less.

## 3.3 Regularization and parameter estimation

In practice, the characteristics of the source distribution, $\Xi$, are not precisely known. Instead, empirical estimates for moment-matching constraints, $\tilde{\mathbf{c}} \triangleq \mathbb{E}_{\tilde{P}_{\text{src}}(x)\tilde{P}(y|x)}[\mathbf{f}(X,Y)]$, are available, but are prone to sampling error. When the constraints of (4) are relaxed using various convex norms, $||\tilde{\mathbf{c}} - \mathbb{E}_{\tilde{P}_{\text{src}}(x)\hat{P}(y|x)}[\mathbf{f}(X,Y)]|| \leq \epsilon$, the RBA classifier is obtained by $\ell_1$- or $\ell_2$-regularized maximum conditional likelihood estimation (Theorem 2) of the dual optimization problem [23, 24],

$$\theta = \underset{\theta}{\operatorname{argmax}} \; \mathbb{E}_{P_{\text{trg}}(x)P(y|x)}\left[\log \hat{P}_\theta(Y|X)\right] - \epsilon \, ||\theta||. \tag{6}$$

The regularization parameters in this approach can be chosen using straight-forward bounds on finite sampling error [24]. In contrast, the sample reweighted approach to learning under sample selection bias [1, 7] also makes use of regularization [9], but appropriate regularization parameters for it must be haphazardly chosen based on how well the source samples represent the target data.

Maximizing this regularized target conditional likelihood (6) appears difficult because target data from $P_{\text{trg}}(x)P(y|x)$ is unavailable. We avoid the sample reweighted approach (2) [1, 7], due to its inaccuracies when facing distributions with large differences in bias given finite samples. Instead, we use the gradient of the regularized target conditional likelihood and only rely on source samples adequately approximating the source distribution statistics (a standard assumption for IID learning):

$$\nabla_\theta \mathbb{E}_{P_{\text{trg}}(x)P(y|x)}[\log \hat{P}_\theta(Y|X)] = \tilde{\mathbf{c}} - \mathbb{E}_{\tilde{P}_{\text{src}}(x)\hat{P}(y|x)}[\mathbf{f}(X,Y)]. \tag{7}$$

Algorithm 1 is a batch gradient algorithm for parameter estimation under our model. It does not require objective function calculations and converges to a global optimum due to convexity [22].

**Algorithm 1** Batch gradient for robust bias-aware classifier learning.

---
**Input:** Dataset $\{(x_i, y_i)\}$, source density $P_{\text{src}}(x)$, target density $P_{\text{trg}}(x)$, feature function $\mathbf{f}(x, y)$, measured statistics $\tilde{\mathbf{c}}$, (decaying) learning rate $\{\gamma_t\}$, regularizer $\epsilon$, convergence threshold $\tau$
**Output:** Model parameters $\theta$

   $\theta \leftarrow \mathbf{0}$
   **repeat**
      $\psi(x_i, y) \leftarrow \frac{P_{\text{src}}(x)}{P_{\text{trg}}(x)} \theta \cdot \mathbf{f}(x_i, y)$ for all: dataset examples i, labels y
      $\hat{P}(Y_i = y | x_i) \leftarrow \frac{e^{\psi(x_i, y)}}{\sum_{y'} e^{\psi(x_i, y')}}$ for all: dataset examples $i$, labels $y$
      $\nabla \mathcal{L} \leftarrow \tilde{\mathbf{c}} - \frac{1}{N} \sum_{i=1}^{N} \sum_{y \in \mathcal{Y}} \hat{P}(Y_i = y | x_i) \, \mathbf{f}(x_i, y)$
      $\theta \leftarrow \theta + \gamma_t (\nabla \mathcal{L} + \epsilon \nabla_\theta ||\theta||)$
   **until** $||\epsilon \nabla_\theta ||\theta|| + \nabla \mathcal{L}|| \leq \tau$
   **return** $\theta$

---

### 3.4 Incorporating expert knowledge and generalizing the reweighted approach

In many settings, expert knowledge may be available to construct the constraint set $\Xi$ instead of, or in addition to, statistics $\tilde{\mathbf{c}} \triangleq \mathbb{E}_{\tilde{P}_{\text{src}}(x) \hat{P}(y|x)}[\mathbf{f}(X, Y)]$ estimated from source data. Expert-provided source distributions, feature functions, and constraint statistic values, respectfully denoted $P'_{\text{src}}(x)$, $\mathbf{f}'(x, y)$, and $\mathbf{c}'$, can be specified to express a range of assumptions about the conditional label distribution and how it generalizes. Theorem 3 establishes that for empirically-based constraints provided by the expert, $\mathbb{E}_{P_{\text{trg}}(x) \hat{P}(y|x)}[\mathbf{f}(X, Y)] = \tilde{\mathbf{c}}' \triangleq \mathbb{E}_{\tilde{P}_{\text{src}}(x) \hat{P}(y|x)}[(P_{\text{trg}}(X)/P_{\text{src}}(X))\mathbf{f}(X, Y)]$, corresponding to strong source-to-target feature generalization assumptions, $P'_{\text{src}}(x) \triangleq P_{\text{trg}}(x)$, reweighted logloss minimization is a special case of our robust bias-aware approach.

**Theorem 3.** *When direct feature generalization of reweighting source samples to the target distribution is assumed, the constraints become* $\mathbb{E}_{P_{\text{trg}}(x) \hat{P}(y|x)}[\mathbf{f}(X, Y)] = \tilde{\mathbf{c}}' \triangleq \mathbb{E}_{\tilde{P}_{\text{src}}(x) \hat{P}(y|x)}\left[\frac{P_{\text{trg}}(X)}{P_{\text{src}}(X)}\mathbf{f}(X, Y)\right]$ *and the RBA classifier minimizes sample reweighted logloss* (2).

This equivalence suggests that *if* there is expert knowledge that reweighted source statistics are representative of the target distribution, then these strong generalization assumptions should be included as constraints in the RBA predictor and results in the sample reweighted approach[1].

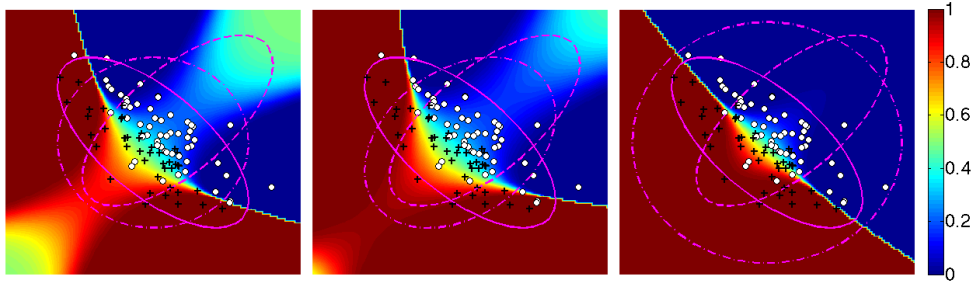

Figure 4: The robust estimation setting of Figure 3 (bottom, right) with assumed Gaussian feature distribution generalization (dashed-dotted oval) incorporated into the density ratio. Three increasingly broad generalization distributions lead to reduced target prediction uncertainty.

Weaker expert knowledge can also be incorporated. Figure 4 shows various assumptions of how widely sample reweighted statistics are representative across the input space. As the generalization assumptions are made to align more closely with the target distribution (Figure 4), the regions of uncertainty shrink substantially.

# 4 Experiments and Comparisons

## 4.1 Comparative approaches and implementation details

We compare three approaches for learning classifiers from biased sample source data: (a) **source logistic regression** maximizes conditional likelihood on the source data, $\max_\theta \mathbb{E}_{\tilde{P}_{\text{src}}(x)\tilde{P}(y|x)}[\log P_\theta(Y|X) - \epsilon||\theta||]$; (b) **sample reweighted target logistic regression** minimizes the conditional likelihood of source data reweighted to the target distribution (2), $\max_\theta \mathbb{E}_{\tilde{P}_{\text{src}}(x)\tilde{P}(y|x)}[(P_{\text{trg}}(x)/P_{\text{src}}(x))\log P_\theta(Y|X) - \epsilon||\theta||]$; and **robust bias-aware classification** robustly minimizes target distribution logloss (5) trained using direct gradient calculations (7). As statistics/features for these approaches, we consider $n^{\text{th}}$ order uni-input moments, e.g., $yx_1, yx_2^2, yx_3^n, \ldots$, and mixed moments, e.g., $yx_1, yx_1x_2, yx_3^2x_5x_6, \ldots$. We employ the CVX package [25] to estimate parameters of the first two approaches and batch gradient ascent (Algorithm 1) for our robust approach.

## 4.2 Empirical performance evaluations and comparisons

We empirically compare the predictive performance of the three approaches. We consider four classification datasets, selected from the UCI repository [6] based on the criteria that each contains roughly 1,000 or more examples, has discretely-valued inputs, and has minimal missing values. We reduce multi-class prediction tasks into binary prediction tasks by combining labels into two groups based on the plurality class, as described in Table 1.

Table 1: Datasets for empirical evaluation

| Dataset | Features | Examples | Negative labels | Positive labels |
|---------|----------|----------|-----------------|-----------------|
| Mushroom | 22 | 8,124 | *Edible* | *Poisonous* |
| Car | 6 | 1,728 | *Not acceptable* | all others |
| Tic-tac-toe | 9 | 958 | *'X' does not win* | *'X' wins* |
| Nursery | 8 | 12,960 | *Not recommended* | all others |

We generate biased subsets of these classification datasets to use as source samples and unbiased subsets to use as target samples. We create source data bias by sampling a random likelihood function from a Dirichlet distribution and then sample source data without replacement in proportion to each datapoint's likelihood. We stress the inherent difficulties of the prediction task that results; label imbalance in the source samples is common, despite sampling independently from the example label (given input values) due to source samples being drawn from focused portions of the input space. We combine the likelihood function and statistics from each sample to form naïve source and target distribution estimates. The complete details are described in Appendix C, including bounds imposed on the source-target ratios to limit the effects of inaccuracies from the source and target distribution estimates.

We evaluate the source logistic regression model, the reweighted maximum likelihood model, and our bias-adaptive robust approach. For each, we use first-order and second-order non-mixed statistics: $x_1^2y, x_2^2y, \ldots, x_K^2y, x_1y, x_2y, \ldots, x_Ky$. For each dataset, we evaluate target distribution logloss, $\mathbb{E}_{\tilde{P}_{\text{trg}}(x)\tilde{P}(y|x)}[-\log \hat{P}(Y|X)]$, averaged over 50 random biased source and unbiased target samples. We employ $\log_2$ for our loss, which conveniently provides a baseline logloss of 1 for a uniform distribution. We note that with exceedingly large regularization, all parameters will be driven to zero, enabling each approach to achieve this baseline level of logloss. Unfortunately, since target labels are assumed not to be available in this problem, obtaining optimal regularization via cross-validation is not possible. After trying a range of $\ell_2$-regularization weights (Appendix C), we find that heavy $\ell_2$-regularization is needed for the logistic regression model and the reweighted model in our experiments. Without this heavy regularization, the logloss is often extremely high. In contrast, heavy regularization for the robust approach is not necessary; we employ only a mild amount of $\ell_2$-regularization corresponding to source statistic estimation error.

We show a comparison of individual predictions from the reweighted approach and the robust approach for the *Car* dataset on the left of Figure 5. The pairs of logloss measures for each of the 50

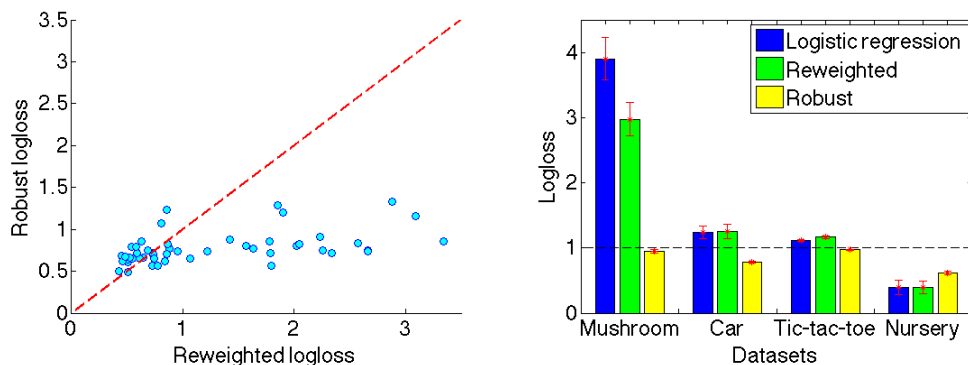

Figure 5: *Left:* Log-loss comparison for 50 source and target distribution samples between the robust and reweighted approaches for the *Car* classification task. *Right:* Average logloss with 95% confidence intervals for logistic regression, reweighted logistic regression, and bias-adaptive robust target classifier on four UCI classification tasks.

sampled source and target datasets are shown in the scatter plot. For some of the samples, the inductive biases of the reweighted approach provide better predictions (left of the dotted line). However, for many of the samples, the inductive biases do not fit the target distribution well and this leads to much higher logloss.

The average logloss for each approach and dataset is shown on the right of Figure 5. The robust approach provides better performance than the baseline uniform distribution (logloss of 1) with statistical significance for all datasets. For the first three datasets, the other two approaches are significantly worse than this baseline. The confidence intervals for logistic regression and the reweighted model tend to be significantly larger than the robust approach because of the variability in how well their inductive biases generalize to the target distribution for each sample. However, the robust approach is not a panacea for all sample selection bias problems; the *No Free Lunch* theorem [26] still applies. We see this with the *Nursery* dataset, in which the inductive biases of the logistic regression and reweighted approaches do tend to hold across both distributions, providing better predictions.

# 5   Discussion and Conclusions

In this paper, we have developed a novel minimax approach for probabilistic classification under sample selection bias. Our approach provides the parametric distribution (5) that minimizes worst-case logloss (Def. 1), and that can be estimated as a convex optimization problem (Alg. 1). We showed that sample reweighted logloss minimization [1, 7] is a special case of our approach using very strong assumptions about how statistics generalize to the target distribution (Thm. 3). We illustrated the predictions of our approach in two toy settings and how those predictions compare to the more-certain alternative methods. We also demonstrated consistent "better than uninformed" prediction performance using four UCI classification datasets—three of which prove to be extremely difficult for other sample selection bias approaches.

We have treated density estimation of the source and target distributions, or estimating their ratios, as an orthogonal problem in this work. However, we believe many of the density estimation and density ratio estimation methods developed for sample reweighted logloss minimization [1, 8, 9, 10, 11, 12, 13] will prove to be beneficial in our bias-adaptive robust approach as well. We additionally plan to investigate the use of other loss functions and extensions to other prediction problems using our robust approach to sample selection bias.

# Acknowledgments

This material is based upon work supported by the National Science Foundation under Grant No. #1227495, *Purposeful Prediction: Co-robot Interaction via Understanding Intent and Goals*.

## Footnotes

[1]Similar to the previous section, relaxed constraints $||\tilde{\mathbf{c}}' - \mathbb{E}_{\tilde{P}_{\text{src}}(x) \hat{P}(y|x)}[\mathbf{f}(X, Y)]|| \leq \epsilon$, are employed in practice and parameters are obtained by maximizing the regularized conditional likelihood as in (6).

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
