[Supplementary Material]

# A  Proofs

*Proof of Theorem 1.* Our proof (and theorem) follows Grünwald and Dawid [5]. The two-player game in Definition 1 can be written as:

$$\min_{\hat{P}(Y|X)\in\Delta}\ \max_{\check{P}(Y|X)\in\Delta\,\cap\,\Xi}\ \mathbb{E}_{P_{\text{trg}}(x)\check{P}(y|x)}[-\log\hat{P}(Y|X)].$$

Assuming the constraint set $\Xi$ is convex and a solution exists on the relative interior of the set, strong duality holds and switching the order of the two players yields a solution with equivalent value:

$$\max_{\check{P}(Y|X)\in\Delta\,\cap\,\Xi}\ \min_{\hat{P}(Y|X)\in\Delta}\ \mathbb{E}_{P_{\text{trg}}(x)\check{P}(y|x)}[-\log\hat{P}(Y|X)].$$

Solving the inner minimization problem assuming that we know $\check{P}(Y|X)$, we get the result that $\hat{P}(Y|X)=\check{P}(Y|X)$. Plugging it into the maximizing problem, the whole problem reduces to:

$$\max_{\hat{P}(Y|X)\in\Delta}\ H_{P_{\text{trg}}(x),\hat{P}(y|x)}(Y|X)\triangleq\mathbb{E}_{P_{\text{trg}}(x)\hat{P}(y|x)}[-\log\hat{P}(Y|X)]$$
$$\text{such that: }\mathbb{E}_{P_{\text{src}}(x)\hat{P}(y|x)}[\mathbf{f}(X,Y)]=\mathbf{c}.$$

$\square$

*Proof of Theorem 2.* The constrained optimization problem in (4) can be written as:

$$\max_{\hat{P}(Y|X)}\ \mathbb{E}_{P_{\text{trg}}(x)\hat{P}(y|x)}[-\log\hat{P}(Y|X)]$$
$$\text{such that: }\mathbb{E}_{P_{\text{src}}(x)\hat{P}(y|x)}[f_k(X,Y)]=c_k,\forall k\in\{1,...,K\}$$
$$\forall x\in\mathcal{X}\colon\mathbb{E}_{\hat{P}(y|x)}[1|X]=1$$
$$\forall x\in\mathcal{X},y\in\mathcal{Y}\colon\hat{P}(y|x)\geq 0.$$

Note that the final constraint is superfluous since the domain of the objective function is the positive real numbers. The Lagrangian associated with this problem is:

$$\mathcal{L}(\hat{P}(y|x),\theta,\lambda)=\mathbb{E}_{P_{\text{trg}}(x)\hat{P}(y|x)}[-\log\hat{P}(Y|X)]+\theta\cdot\left(\mathbb{E}_{P_{\text{src}}(x)\hat{P}(y|x)}[\mathbf{f}(X,Y)]-\mathbf{c}\right)$$
$$+\sum_{x\in\mathcal{X}}\lambda(x)\left[\mathbb{E}_{\hat{P}(y|x)}[1|X]-1\right],$$

where $\theta$ and $\lambda(x)$ are the Lagrangian multipliers[2]. According to strong Lagrangian duality (assuming a solution on the relative interior of the constraint set),

$$\max_{\hat{P}(y|x)\in\Delta}\ \min_{\theta,\lambda(x)}\mathcal{L}(\hat{P}(y|x),\theta,\lambda(x))=\min_{\theta,\lambda(x)}\ \max_{\hat{P}(y|x)\in\Delta}\mathcal{L}(\hat{P}(y|x),\theta,\lambda(x)).$$

So, assuming a fixed $\theta$ and $\lambda(x)$, the internal maximization problem can be solved first. Taking the partial derivative with respect to the conditional probability of a specific $y$ and $x$, $\hat{P}(y|x)$,

$$\frac{\partial}{\partial\hat{P}(y|x)}\mathcal{L}(\hat{P}(y|x),\theta,\lambda)=-P_{\text{trg}}(x)\log\hat{P}(y|x)-P_{\text{trg}}(x)+P_{\text{src}}(x)\theta\cdot\mathbf{f}(x,y)+\lambda(x),$$

setting it equal to zero, $\frac{\partial}{\partial\hat{P}(y|x)}\mathcal{L}(\hat{P}(y|x),\theta,\lambda(x))=0$, and solving, we obtain:

$$\log\hat{P}(y|x)=-1+\frac{P_{\text{src}}(x)}{P_{\text{trg}}(x)}\theta\cdot\mathbf{f}(x,y)+\frac{\lambda(x)}{P_{\text{trg}}(x)}.$$

Therefore, we conclude:

$$\hat{P}(y|x)=e^{\frac{P_{\text{src}}(x)}{P_{\text{trg}}(x)}\theta\cdot\mathbf{f}(x,y)-1+\frac{\lambda(x)}{P_{\text{trg}}(x)}}.$$

We analytically solve the normalization Lagrange multiplier terms,

$$\lambda(x) = P_{\text{trg}}(x)\left(-\log\sum_{y'\in\mathcal{Y}} e^{\frac{P_{\text{src}}(x)}{P_{\text{trg}}(x)}\theta\cdot\mathbf{f}(x,y')} + 1\right),\tag{8}$$

yielding the conditional probability distribution of labels (with $Z_\theta(x) \triangleq \sum_{y'\in\mathcal{Y}} e^{\frac{P_{\text{src}}(x)}{P_{\text{trg}}(x)}\theta\cdot\mathbf{f}(x,y')}$):

$$\hat{P}(y|x) = \frac{e^{\frac{P_{\text{src}}(x)}{P_{\text{trg}}(x)}\theta\cdot\mathbf{f}(x,y)}}{Z_\theta(x)} = \frac{e^{\frac{P_{\text{src}}(x)}{P_{\text{trg}}(x)}\theta\cdot\mathbf{f}(x,y)}}{\sum_{y'\in\mathcal{Y}} e^{\frac{P_{\text{src}}(x)}{P_{\text{trg}}(x)}\theta\cdot\mathbf{f}(x,y')}}.\tag{9}$$

Plugging the expression back into the Lagrangian and solving the outer minimization problem, we obtain

$$\mathcal{L}(\theta) = \mathbb{E}_{P_{\text{trg}}(x)\hat{P}(y|x)}\left[\log Z_\theta(X) - \frac{P_{\text{src}}(X)}{P_{\text{trg}}(X)}\theta\cdot\mathbf{f}(X,Y)\right] + \theta\cdot\left(\mathbb{E}_{P_{\text{src}}(x)\hat{P}(y|x)}[\mathbf{f}(X,Y)] - \mathbf{c}\right)$$
$$= \mathbb{E}_{P_{\text{trg}}(x)}[\log Z_\theta(X)] - \theta\cdot\mathbf{c}.$$

Thus, the optimal Lagrangian parameters from the dual optimization problem are: $\theta = \arg\min_\theta \mathbb{E}_{P_{\text{trg}}(x)}[\log Z_\theta(X)] - \theta\cdot\mathbf{c}$.

Given $c_k = \mathbb{E}_{P_{\text{src}}(x)P(y|x)}[f_k(X,Y)]$, the parameter estimation can be regarded as maximizing the expectation of the log-likelihood over target distribution under $\hat{P}_\theta(y|x) = \frac{e^{\frac{P_{\text{src}}(x)}{P_{\text{trg}}(x)}\theta\cdot\mathbf{f}(x,y)}}{Z_\theta(x)}$:

$$\min_\theta \mathcal{L}(\theta) = \min_\theta\left(\mathbb{E}_{P_{\text{trg}}(x)}[\log Z_\theta(X)] - \theta\cdot\mathbf{c}\right)$$
$$= \min_\theta\left(\mathbb{E}_{P_{\text{trg}}(x)}[\log Z_\theta(X)] - \theta\cdot\mathbb{E}_{P_{\text{src}}(x)P(y|x)}[\mathbf{f}(X,Y)]\right)$$
$$= \min_\theta\left(\mathbb{E}_{P_{\text{trg}}(x)}[\log Z_\theta(X)] - \theta\cdot\mathbb{E}_{P_{\text{trg}}(x)P(y|x)}\left[\frac{P_{\text{src}}(X)}{P_{\text{trg}}(X)}\mathbf{f}(X,Y)\right]\right)$$
$$= \max_\theta \mathbb{E}_{P_{\text{trg}}(x)P(y|x)}\left[\log\left(\frac{e^{\frac{P_{\text{src}}(X)}{P_{\text{trg}}(X)}\theta\cdot\mathbf{f}(X,Y)}}{Z_\theta(X)}\right)\right]$$
$$= \max_\theta \mathbb{E}_{P_{\text{trg}}(x)P(y|x)}[\log\hat{P}_\theta(Y|X)].$$

We use the gradient of the exact target likelihood to estimate the parameters. Taking the derivative with respect to $\theta$, we obtain

$$\frac{\partial}{\partial\theta_k}\mathbb{E}_{P_{\text{trg}}(x)P(y|x)}[\log\hat{P}_\theta(Y|X)] = \frac{\partial}{\partial\theta_k}\left(\theta\cdot\mathbf{c} - \mathbb{E}_{P_{\text{trg}}(x)}[\log Z_\theta(X)]\right)$$
$$= c_k - \mathbb{E}_{P_{\text{trg}}(x)}\left[\sum_{y\in\mathcal{Y}}\frac{e^{\frac{P_{\text{src}}(X)}{P_{\text{trg}}(X)}\theta\cdot\mathbf{f}(X,y)}}{Z_\theta(X)}\frac{P_{\text{src}}(X)}{P_{\text{trg}}(X)}f_k(X,y)\right]$$
$$= c_k - \mathbb{E}_{P_{\text{trg}}(x)\hat{P}(y|x)}\left[\frac{P_{\text{src}}(X)}{P_{\text{trg}}(X)}f_k(X,Y)\right]$$
$$= c_k - \mathbb{E}_{P_{\text{src}}(x)\hat{P}(y|x)}[f_k(X,Y)].$$

So if we use a gradient ascent method to estimate the parameter, the gradient in each step for $\theta_k$ is $c_k - \mathbb{E}_{P_{\text{src}}(x)\hat{P}(y|x)}[f_k(X,Y)]$. □

*Proof of Theorem 3.* Assuming $P'_{\text{src}}(x) = P_{\text{trg}}(x)$ for feature expectations in the constraints of (4), and following the same approach as the proof of Theorem 2, we obtain the form of the RBA classifier in this case, which is the same as logistic regression, $\hat{P}(y|x) = \frac{e^{\theta\cdot\mathbf{f}(x,y)}}{Z'(x)}$, with $Z'_\theta(x) = \sum_{y'\in\mathcal{Y}} e^{\theta\cdot\mathbf{f}(x,y')}$.

Plugging this parametric form into the Lagrangian and solving the minimization problem, applying that $\tilde{c}'_k = \mathbb{E}_{\tilde{P}_{\text{src}}(x)\hat{P}(y|x)}\left[\frac{P_{\text{trg}}(X)}{P_{\text{src}}(X)}f_k(X,Y)\right]$, the problem becomes minimizing the following:

$$\mathcal{L}(\theta) = \mathbb{E}_{P_{\text{trg}}(x)\hat{P}(y|x)}\left[Z'_\theta(X) - \theta \cdot \mathbf{f}(X,Y)\right]$$

$$+ \theta \cdot \left(\mathbb{E}_{P_{\text{trg}}(x)\hat{P}(y|x)}[\mathbf{f}(X,Y)] - \mathbb{E}_{\tilde{P}_{\text{src}}(x)\tilde{P}(y|x)}\left[\frac{P_{\text{trg}}(X)}{P_{\text{src}}(X)}\mathbf{f}(X,Y)\right]\right)$$

$$= \mathbb{E}_{P_{\text{trg}}(x)}[\log Z'_\theta(X)] - \theta \cdot \tilde{\mathbf{c}}'. \tag{10}$$

The gradient of (10) is $\mathbb{E}_{P_{\text{trg}}(x)\hat{P}(y|x)}[\mathbf{f}(X,Y)] - \tilde{\mathbf{c}}' = \mathbb{E}_{P_{\text{src}}(x)\hat{P}(y|x)}\left[\frac{P_{\text{trg}}(X)}{P_{\text{src}}(X)}\mathbf{f}(X,Y)\right] - \tilde{\mathbf{c}}' \approx \mathbb{E}_{\tilde{P}_{\text{src}}(x)\hat{P}(y|x)}\left[\frac{P_{\text{trg}}(X)}{P_{\text{src}}(X)}\mathbf{f}(X,Y)\right] - \tilde{\mathbf{c}}'$. Constraint slack and dual regularization can be applied to allow for the noise from finite sample approximation, as described in §3.3. We omit these in the interest of clarity and brevity. The sample reweighted logloss minimization problem assumes $\hat{P}_\theta(y|x) = \frac{e^{\theta \cdot \mathbf{f}(x,y)}}{\sum_{y' \in \mathcal{Y}} e^{\theta \cdot \mathbf{f}(x,y')}}$ and minimizes the following:

$$ll(\theta) = \mathbb{E}_{P_{\text{trg}}(x)P(y|x)}[-\log \hat{P}_\theta(Y|X)]$$

$$\approx \mathbb{E}_{\tilde{P}_{\text{src}}(x)\tilde{P}(y|x)}\left[-\frac{P_{\text{trg}}(X)}{P_{\text{src}}(X)}\log \hat{P}_\theta(Y|X)\right]$$

$$= \mathbb{E}_{\tilde{P}_{\text{src}}(x)}\left[\frac{P_{\text{trg}}(X)}{P_{\text{src}}(X)}\log Z'_\theta(X)\right] - \mathbb{E}_{\tilde{P}_{\text{src}}(x)\tilde{P}(y|x)}\left[\frac{P_{\text{trg}}(X)}{P_{\text{src}}(X)}\theta \cdot \mathbf{f}(X,Y)\right]$$

$$= \mathbb{E}_{\tilde{P}_{\text{src}}(x)}\left[\frac{P_{\text{trg}}(X)}{P_{\text{src}}(X)}\log Z'_\theta(X)\right] - \theta \cdot \tilde{\mathbf{c}}'. \tag{11}$$

The approximation in the second step is based on the sample reweighting assumption (2) . The gradient of (11) is $\mathbb{E}_{\tilde{P}_{\text{src}}(x)\hat{P}(y|x)}\left[\frac{P_{\text{trg}}(X)}{P_{\text{src}}(X)}\mathbf{f}(X,Y)\right] - \tilde{\mathbf{c}}'$, which is the same with the gradient of (10). Therefore, the two approaches are equivalent in this special case. □

## B   Additional Synthetic Experiment Results and Experimental Details

We include additional synthetic experiments that did not fit into the main paper due to space constraints. Figure 6 shows the predictions of each approach with higher-order statistics using the same sample as Figure 3. The robust prediction using fourth moments does not vary significantly from the second-moment predictions.

Figure 6: The prediction setting and data sample of Figure 3 with fourth-order mixed moments.

Figure 7 shows the same first moment and second moment predictions of Figure 3 for a different data sample.

Note that for logistic regression and the reweighted approach, the predictions where the source density is greater are very similar to the predictions given the previous data sample. However, the predictions where there is not much data—where inductive extrapolation is occurring—can vary significantly between samples. In contrast, the predictions of the robust approach are more consistent across data samples.

Figure 7: The same prediction conditions as Figure 3 using a different data sample.

For our feature generalization experiments (Figure 4), we assume all features generalize to the indicated generalization distributions. We use sample reweighting from the source distribution to the generalization distribution to obtain the constraint constants for each feature $k$,

$$\tilde{c}'_k = \mathbb{E}_{\tilde{P}_{\mathrm{src}}(x)} \left[ \frac{P^k_{\mathrm{gen}}(x)}{P_{\mathrm{src}}(x)} f_k(X, Y) \right].$$

## C   Experimental Details for UCI Classification Datasets

We sub-sample each original dataset to create biased source datasets using the following procedure:

1. Randomly split half of the dataset into a training set and half into a testing set.

2. For each input dimension, independently sample $P_{\mathrm{src}}(x_k)$ uniformly from the $(|\mathcal{X}_k| - 1)$-simplex (i.e., a Dirichlet$(1, \ldots, 1)$ distribution).

3. Compute $P_{\mathrm{src}}(x_i)$ for each example in the training datasets.

4. Sub-sample $N$ examples from the training distributions in proportion to $P_{\mathrm{src}}(x_i)$ to form $\tilde{P}_{\mathrm{src}}(x)$.

5. Set $P_{\mathrm{src}}(x_k) \leftarrow \alpha P_{\mathrm{src}}(x_k) + (1 - \alpha)\tilde{P}_{\mathrm{src}}(x_k)$ for each input dimension $k$.

We incorporate the fifth step to address datasets with very low likelihood around the sampled source probability distributions. In these datasets, the empirical distribution would deviate substantially from the initial source distributions otherwise. We use $N = 100$ and $\alpha = 0.5$ in our experiments.

Our source distributions make a strong independence assumption, $P(x) = \prod_{k=1}^{K} P(x_k)$. We limit the negative influence of this naïve independence assumption by bounding the source-target probability ratio to $[0.0001, 1]$. Without these bounds, we encounter a large number of target samples that should occur less than "one in a billion" times in our samples of 100 examples.

For the *Mushroom* dataset, we omitted the *stalk-root* feature in our experiments due to it having missing values for some instances.

We considered a range of regularization weights and report the best one for each dataset. Table 2 lists the best regularization weights employed for each dataset and approach.

Table 2: Regularization Weight for Different Datasets

| Dataset | Logistic Regression | Reweighted | Robust |
|---|---|---|---|
| Mushroom | 5 | 10 | 0.02 |
| Car | 0.5 | 0.2 | 0 |
| Nursery | 0.2 | 0.1 | 0.02 |
| Tic-tac-toe | 0.5 | 0.5 | 0 |

Using the best regularizatin weights is generous to the logistic regression and reweighted approaches, as their regularization weights are based on how well their inductive biases hold in the target distribution. This is unknown in the sample selection bias setting. In contrast, approximation error rates for the source distribution statistics guide appropriate regularization parameters for the robust approach. As noted in Section 4.2, extremely large regularization can be employed to reduce (or increase) the logloss to 1, but then nothing is learned.

## D  Additional UCI Classification Dataset Results

Figure 8 shows an additional set of experimental results on UCI datasets when the target distribution is also sampled according to the same biased sampling procedure as the source distribution (with a different "seed" likelihood function).

Figure 8: *Left:* Log-loss comparison for 50 biased source and biased target distribution samples between the robust and reweighted approaches for the *Car* classification task. *Right:* Average logloss with 95% confidence intervals for logistic regression, reweighted logistic regression, and bias-adaptive robust target classifier on four UCI classification tasks.

The results of predicting biased target datasets are qualitatively similar to the results of predicting unbiased target datasets (Figure 5).

## Footnotes

[2] For continuous input spaces $\mathcal{X}$, the sum over $x\in\mathcal{X}$ is replaced by an integral over $x\in\mathcal{X}$, but the resulting distribution form is unchanged.