[Reviews · NeurIPS 2014]

Submitted by Assigned_Reviewer_4

Paper ID: 39
Title: Robust Classification Under Sample Selection Bias

NOTE: Due to the short reviewing time and out of fairness to the other papers I was reviewing, I DID NOT do more than glance over the supplementary material. In most cases, I therefore did not verify that results claimed by the authors are correct, but instead checked that they are plausible.

Summary: This paper is about how to adjust a classifier when the training set is not representative of the test set; a canonical example is active learning, but the problem can also appear in recalibration. The goal is to find an effective method in the finite-sample regime, since most results are asymptotics. The author(s) take a minimax approach.

Quality, clarity, originality and significance: This work takes a novel approach to address an important problem in the practical application of classifiers. The manuscript is generally well-written and clear, but I have some comments about works that seem related to this problem. Some imprecision in the mathematical description was confusing.

Recommendation: Overall, I believe this is a nice contribution which will be appreciated by the NIPS community, assuming that the manuscript can be clarified appropriately.

Comments: (comments are numbered to allow for easy reference in the rebuttal)

1) There are some works on recalibrating classifiers to adjust for different populations (a Google search turned up some papers from IJCAI, CVPR, KDD, and other venues). Is there a connection between this problem and that literature that may be relevant?

2) The formulation in (1) assumes the alphabets for X and Y are both discrete, but this is not what happens in the experiments. It may be better to express (1) as an expectation rather than writing it out explicitly. Otherwise the generality is lost, since one has to replace sums by integrals.

3) After Definition 1 it may be appropriate to comment on why matching the constraints *exactly* is even relevant in this setting (this is relaxed later but at the moment Definition 1 seems already like a bad idea).

4) In 2.2, I was not sure what the authors meant by "assuming that source distribution statistics generalize more broadly than the source distribution itself may be justifiable and useful." What does expert knowledge have to do with this? I was rather confused by this motivation.

5) By introducing some slack in the constraints the estimator seems somewhat "similar" to things one sees in robust statistics. Is there a connection here?

6) Somehow the form of the estimator in Theorem 2 comes out of the blue. Is there a way to explain some of the intuition behind it before introducing it?

7) What does \approx mean in (8)? That seems very imprecise.

8) In 2.5, this goes back to 2) -- densities have thus far not appeared in the story and suddenly it seems all variables are continuous...

9) In Algorithm 1, how does gamma relate to epsilon in the error bounds in 2.4?

10) In 3.1, are the other methods from other papers on this problem? If so, please provide citations.

11) In general, in the experiments section, using some notation that had been previously established to connect the plots back to the preceding theory would be extremely helpful.

12) Just as a thought, how would increasing the training set size help in Figures 2-3?

Small things:
- Perhaps before Definition 2 the c' variables could be actually defined, e.g. using \stackrel{\Delta}{=} or some other macro.
- check the referenes -- there are some typos (c.f. Bickel's first name in

Please summarize your review in 1-2 sentences: The authors propose a new method for handling mismatch between the training and test distributions when training a classifier. Experiments illustrate the benefit of the method over competing approaches

Quality Score (1-10): 7

Impact Score (1-2): 2

Confidence (1-5): 3

COMMENTS AFTER AUTHOR FEEDBACK: The authors have proposed a rather extensive set of revisions to address the feedback from the reviewers. With those changes I think the paper may indeed be in the top 50% of NIPS papers. I am therefore revising my score upward to an 8.

Quality Score (1-10): 8
Summary: The authors propose a new method for handling mismatch between the training and test distributions when training a classifier. Experiments illustrate the benefit of the method over competing approaches

Submitted by Assigned_Reviewer_15

This paper proposes a new method for adapting the classifier learned from a source distribution to a target distribution (e.g., training data have significant sample selection biases).

This paper first proposes a minimax formulation which is then equivalent to maximizing a entropy with respect to the target distribution. Then the target distribution (is unknow and hence) is replaced by the generalization distributions. This is because of the belief that the k-th statistic of the source distribution generalizes to additional portion of the input space. Then, the new formulation is proposed using sample rewriting only from the source distribution to the generalization distribution in eq(9), which is then compared with existing methods [1,4] as shown in eq(8) which uses the target-source reweighting. It would be better to define clearly \hat{P}, \tilde{P} and P in the different formulations.

The method appears to be original, and may have noticeable impact on the domain adaptation field. The paper is well written although has several places that need to be clarified, such as the definitions of notation. This paper does not cite or discuss quite a few other domain adaptation papers that deal with similar problems, such as “Reshaping Visual Datasets for Domain Adaptation”, NIPS 2013. Those methods used very different approaches from this paper to adapt the classifiers when the target distribution is different from the training distribution.

The simulations are quite useful to understand the nature of the problem and the behavior of the proposed method. For the real-world datasets, how was the performance evaluated? Evaluated against the target samples or the original classification datasets? In the experiments, the first-order and second-order statistics were used. In general, what statistics should be considered (as generalization distributions) for the proposed approach? Can other domain adaptation methods be compared with the proposed approach other than the two reweighting based methods?
Summary: The paper presents an interesting idea of using generalization distributions to adapt classifiers from source distribution to target distribution. The paper does not discuss a few other state-of-the-art methods for comparison.

Submitted by Assigned_Reviewer_45

This paper addresses the problem of learning under a distribution
shift, ie, when the training and test data are not sampled from the
same distribution. The proposed approach is a minimization of a loss
over the target distribution, under a worst case scenario for the
actual distribution generating the training data. The problem is shown
to be equivalent to an unconstrained minimization of a different
objective. The gradient of this objective can be computed efficiently
and without resorting to sample reweighting.

The tackled problem is relevant and original to the best of my
knowledge: reweighting techniques can indeed fail when limited data is
available, and it is interesting to consider a technique which works
around the problem. The paper also seems to be technically sound.

My main concern is on clarity. I didn't find the paper self
sufficient. Theorem 2 in particular is stated in vague terms, and I
had to use the supplementary material to understand what was really
said. Even there, I found some statements lacked specificity. For
example, the original problem (2, 3) is stated in terms of an
optimization over a set of distributions. In Theorem 2, it is said to
be equivalent to a parametric problem. In supplementary material, the
dual of (3) is computed by taking the derivative of the objective with
respect to the distribution, without specifying which type of
derivative is used. Similarly, it is difficult to understand what is
done for the experiments on UCI data without reading the supplementary
material.

I am also not entirely convinced by the experiments. Experiments on
synthetic data are used to show the the proposed approach is better at
representing uncertainty in low density regions. It would have been
useful to discuss settings where the proposed approach clearly
outperformed existing methods. The experiments on UCI data are
favorable to the proposed approach. However in most cases it is only
marginally better than classification using a uniform conditional
distribution. Is it possible that the constructed sampling
distribution shifts made the problem too difficult?

[UPDATE] I thank the authors for their detailed answers.
Summary: Novel and original approach for learning under a dsitribution shift. The paper looks technically sound, but lacks clarity and is not self sufficient.
Author Feedback
Author rebuttal: Thank you Assigned_Reviewer_15 (R1), Assigned_Reviewer_4 (R2), and Assigned_Reviewer_45 (R3) for your detailed reading and useful comments. Since the primary concerns (clarity, experiments, and related work) are shared among reviewers, we discuss each in turn.

CLARITY. Our revision clarifies our approach and contribution as follows:

-- We clarify that sample reweighting is the most prominent general-purpose covariate shift technique. It attempts to debias the source-target data difference, but it only succeeds in general by debiasing asymptotically (i.e., (8) holds with equality only with an infinite number of samples for \tilde{P} on the right-hand side (R2.7)). For any finite number of samples under general distribution shift, the approximation (8) can be arbitrarily bad (see Cortes et al. NIPS 2010 for analyses).

-- We revise Thm. 1 so that exact statistics from the source distribution are used in the rhs of the constraint rather than source samples and explain that this provides an upper bound on the target logloss (R2.3). This is later relaxed to use source samples to form the constraint when discussing regularization (12), which is explained in more detail. We explicitly add regularization into Alg. 1 (which now solves (10)). The regularization is unrelated to the learning rate (R2.9) and addresses finite source sample error of \tilde{c} rather than data outliers as in robust statistics (R2.5).

-- We de-emphasize “incorporating expert knowledge” in our revision. Our primary goal with this extension was to show that under very strong “expert knowledge” assumptions, our robust approach is equivalent to reweighted learning (Thm. 3). However, we agree that it made the derivation overly complicated (R3) and was not explained clearly (R2.4). We explain it more intuitively now as allowing an expert to choose the P_src, f, and c components of the constraints in e.g., (3) (rather than basing them on actual source data) (R2.4) and clarify that expert knowledge is employed only in the experiments of Fig. 3. Other experiments use (10) (R1).

-- We focus our derivation on the source statistic-based constrained optimization (3) rather than the expert-provided constraints (5). This yields the simpler distribution form (10) as the main result for Thm 2. Our revision explains the steps of the method of Lagrange multipliers and duality in more detail (R2.6, R3), including strong Lagrangian duality (max_P min_\theta = min_\theta max_P) and using the partial derivative of each \hat{P}(y|x) (R3) to solve the internal maximization in the dual (given the Lagrangian multipliers, \theta). Lagrange multipliers associated for each statistic expectation constraint form the parameters of (10). The proof’s Lagrangian is corrected to also be a function of the set of \hat{P}(y|x) in our revision. We believe this significantly clarifies the approach overall by streamlining its description.

-- We integrate the synthetic experiments of Sec. 3.2 into the method section to better describe and motivate/integrate the approach (R2.11).

-- We fix/explain notation (R1, R2.2, R2.8) and add citations (R2.10) as suggested.

EXPERIMENTS. Existing results use a biased target sample (R1). We agree that an unbiased target sample is a more compelling prediction task (R1,R3).

-- We have conducted additional experiments with an IID target sample (R1,R3). The results are qualitatively similar to the biased-target results.

______________LR__RW__Robust

MUSHROOM___3.90 2.98 0.95

CAR__________1.24 1.25 0.78

TIC-TAC-TOE__1.12 1.17 0.97

NURSERY_____0.40 0.40 0.61

-- Given the performance of other methods, being “marginally” better than “I don’t know” predictions (logloss of 1) with statistical significance is challenging for these tasks (R3). One might hypothesize based on the aggregate results that whenever our method predicts well, reweighted will be even better, but Fig. 4 left clearly shows this is not the case. Further, predictors that appropriately answer with “I don’t know” are exactly what is needed in real systems where those predictions might e.g., drive decision-making or active learning.

-- We report the results of other methods using the best choice of regularization. This is extremely favorable (R3) to these methods since (unlike our approach) this regularization cannot simply be chosen based on source statistic sampling error bounds; a reweighted cross-validation procedure would be required that is equally susceptible to optimistic extrapolation errors.

-- R2.12: In contrast with IID learning, the reweighted target loss estimate does not converge linearly with more samples. So simply increasing the number of samples (short of infinite samples) does not solve general covariate shift problems with the same data efficiency as IID learning.

RELATED WORK

-- We have added a related work section to discuss more broadly how our method is related to other approaches to sample selection bias/domain adaptation/transfer learning.

-- We clarify that many methods in domain adaptation assume that some target data is labeled (e.g., Hal Duame’s work) or do not make the covariate shift assumption (e.g., more general transfer learning tasks) (R1), making applicability in our experiments impossible, or requiring domain expertise to e.g., restrain the hypothesis complexity between source and target data.

-- Methods that leverage additional knowledge (e.g., target distribution marginals for calibration (R2.1)) or structure in the data (e.g., mixtures of domains for “Reshaping Visual Datasets” (R1)) can be employed to improve our robust approach by adding corresponding constraints. We contend that comparisons with these methods, which can orthogonally improve both our approach and sample reweighting, are not warranted (R1), but do serve as important and interesting future work.